# Lack of Informations about COVID-19 Vaccine: From Implications to Intervention for Supporting Public Health Communications in COVID-19 Pandemic

**DOI:** 10.3390/ijerph18116141

**Published:** 2021-06-07

**Authors:** Silva Guljaš, Zvonimir Bosnić, Tamer Salha, Monika Berecki, Zdravka Krivdić Dupan, Stjepan Rudan, Ljiljana Majnarić Trtica

**Affiliations:** 1Department of Radiology, University Hospital Center Osijek, 31 000 Osijek, Croatia; zdravka.krivdic@gmail.com; 2Faculty of Medicine, Josip Juraj Strossmayer University of Osijek, 31 000 Osijek, Croatia; tsalha007@gmail.com (T.S.); m.berecki@gmail.com (M.B.); 3Faculty of Dental Medicine and Health, Josip Juraj Strossmayer University of Osijek, 31 000 Osijek, Croatia; stjepan.rudan@gmail.com; 4Department of Internal Medicine, Family Medicine and the History of Medicine, Faculty of Medicine, Josip Juraj Strossmayer University of Osijek, 31 000 Osijek, Croatia; zbosnic191@gmail.com (Z.B.); ljiljana.majnaric@gmail.com (L.M.T.); 5Department of Teleradiology and Arteficial Intelligence, Health Center Osijek-Baranja County, 31 000 Osijek, Croatia; 6Department of Public Health, Faculty of Dental Medicine and Health, Josip Juraj Strossmayer University of Osijek, 31 000 Osijek, Croatia

**Keywords:** COVID 19, vaccination, health information, health communication, preventive measures

## Abstract

Lack of knowledge and mistrust towards vaccines represent a challenge in achieving the vaccination coverage required for population immunity. The aim of this study is to examine the opinion that specific demographic groups have about COVID-19 vaccination, in order to detect potential fears and reasons for negative attitudes towards vaccination, and to gain knowledge on how to prepare strategies to eliminate possible misinformation that could affect vaccine hesitancy. The data collection approach was based on online questionnaire surveys, divided into three groups of questions that followed the main postulates of the health belief theory—a theory that helps understanding a behaviour of the public in some concrete surrounding in receiving preventive measures. Ordinary least squares regression analyses were used to examine the influence of individual factors on refusing the vaccine, and to provide information on the perception of participants on the danger of COVID-19 infection, and on potential barriers that could retard the vaccine utility. There was an equal proportion of participants (total number 276) who planned on receiving the COVID-19 vaccine (37%), and of those who did not (36.3%). The rest (26.7%) of participants were still indecisive. Our results indicated that attitudes on whether to receive the vaccine, on how serious consequences might be if getting the infection, as well as a suspicious towards the vaccine efficacy and the fear of the vaccine potential side effects, may depend on participants’ age (<40 vs. >40 years) and on whether they are healthcare workers or not. The barriers that make participants‘ unsure about of receiving the vaccine, such as a distrust in the vaccine efficacy and safety, may vary in different socio-demographic groups and depending on which is the point of time in the course of the pandemic development, as well as on the vaccine availability and experience in using certain vaccine formulas. There is a pressing need for health services to continuously provide information to the general population, and to address the root causes of mistrust through improved communication, using a wide range of policies, interventions and technologies.

## 1. Introduction

The Coronavirus disease 19 (COVID-19) pandemic, named severe acute respiratory syndrome coronavirus 2 (SARS-CoV-2), is the defining global health crisis of our time which affected public health, social life, economic strategies and political efficacy [1]. This global phenomenon was firstly identified in Wuhan, China, and quickly spread to now affecting more than 151 millions of people worldwide. Even though it was anticipated that the COVID-19 pandemic could be controlled by practicing social distancing and wearing facial masks, efforts were quickly focused on the new antiviral drugs and an effective vaccine, with the main goal of preventing virus transmission.

Regardless of COVID-19, disease prevention by vaccination is considered one of the greatest successes of public health medicine, which efficacy was confirmed in reducing the global morbidity and mortality of some serious diseases, such as polio, measles, rabies, typhus and some others [2]. Vaccines work by protecting the vaccinated individuals and reducing the possibility of disease transmission in populations with high vaccination coverage, through global immunity [3]. An ongoing public acceptance is required to maintain herd immunity, prevent outbreaks of vaccine preventable illnesses and ensure adoption of novel vaccines. This means that global collaboration and willingness to receive vaccines among the population play a major role [4]. A public concern and an increasing influence of the anti-vaccination movements against globally accepted vaccine programs, are the major problems and a topic of many discussions and scientific papers [5,6]. Regarding the COVID-19 infection, numerous studies were followed in which researchers studied a number of modifications of the COVID-19 vaccine and used a variety of technology platforms, which finally has led to the production of several useful COVID-19 vaccine options. Even though the situation with COVID-19 pandemic is specific and different from previous needs for mass vaccination, in terms of the spread of the pandemic all over the globe, many undetermined factors that influence this spread and the number of experimental vaccines that are now available—the public concern regarding vaccination is always about usefulness and safety of vaccines [7,8,9].

Amongst the barriers to universal vaccination, misinformation regarding the benefits, medicinal composition and adverse effects of vaccination limits patient understanding, and the increase in antivaccination movement was mainly caused by believes that vaccinations do more harm than good to children, especially in case of connecting vaccination with autism [10]. Furthermore, disappearance of some dangerous infectious diseases because of vaccination reduces motivation in the general population to take vaccine under excuse that diseases have been eradicated [11]. A reason for not taking vaccine, especially important in COVID-19 pandemic, is a view that vaccination is not necessary because good hygienic measures could substitute the need for vaccine [12]. Taught by past experience on the effectiveness of vaccination in combating many serious infectious diseases, great hopes were placed in vaccine development and vaccination as the most important epidemiological measure, but potential side effects and questionable efficacy in completely eradicating the infection that occur now that the vaccine is available emphasise the difference between this situation and all previous ones, putting us in a position od constantly adjusting our opinions and attitudes to new information.

Now in the age of modern technology, growing interest has emerged in the role of interactive social media in public health promotion. In contrast to traditional media, social media allow individuals to rapidly create and share content globally without editorial oversight. While this could be very useful as a way of quickly spreading correct information by medical authorities, this also challenge medical and public health authorities, giving patients a wealth of misinformation and anecdotal evidence, and encourages them to participate more actively in medical decision-making, which can have potentially dangerous consequences for the public [13,14,15,16]. This misinformation and unsubstantiated rumours regarding COVID-19 and potential vaccination against COVID-19 have already begun emerging on social media platforms [17,18]. Under the influence of misinformation, the opinion of the younger part of the population can change, so honest reporting of epidemiological and scientific facts should suppress the impact of such information on the formation of public opinion. Social networks should be used in order to inform the public as accurately as possible [19].

The health behaviour and decisions of everyone are influenced by demographically specific social, political and cultural factors, especially in terms of customs, beliefs and opinions of family members and colleagues, as well as personal experiences and current health status [20,21]. The health belief theory, developed in 1950’s, helps us understanding the reasons why some socio-demographic groups resist to receive some preventive measures, including the hesitancy towards vaccination, by revealing the relationships between the self-perceived costs and benefits if one decides to receive some preventive measure (i.e., vaccine). According to the health belief model, people are more receptive to optional vaccination if they believe: a disease is a dangerous condition, they are susceptible to a disease, that vaccination would benefit them by reducing their risk of developing a disease or, at least, serious complications of a disease and that benefit of vaccination overcomes its potential risks—which all greatly depends on (mis)information they receive [22,23]. A study on human influenza virus vaccination in risk groups, by Cheney and John, found that people who are willing to receive the influenza vaccine have different attitudes towards their health risks, and their decisions are strongly influenced by the opinions of family and physicians [24]. This shows that the decision to vaccinate is a dynamic process, subject to the influences of personal beliefs, experiences and social circumstances in which public media have a crucial role. We used the concepts of the health belief theory to monitor the situation regarding self-perception of people in the general population towards COVID-19 vaccination, in the time point before the vaccination has started, and when the vaccine has been rapidly developing around the world. As expected, there was a significant part of respondents who were indecisive regarding vaccination. The public health workers have the task to understand the reasons for this hesitancy and to provide a support in making a decision. The attitude of the general population towards the COVID-19 vaccine, under influence of many factors, has changed from the beginning of the pandemic. Palamenghi L. et al. reported decrease in willingness to vaccinate and a lower trust in science in August 2020. In Italy, in the period between the lockdown and easing of pandemic measures. They pointed to the fact that if people remain to the opinion that they should not receive the vaccine, the rest of the population willing to receive the vaccine will not be enough to prevent the spread of COVID-19 infection [25]. In order to achieve population immunity, 55% to 82% of the population must receive the vaccine or be exposed to the virus [26]. To achieve this, it is necessary for public health institutions to gain the trust of the public and for mutual cooperation to be established through dialogue between the public and health workers [27]. The attitude of the general population towards the COVID-19 vaccine, under influence of many factors, has changed from the beginning of the pandemic. From initial fear and suspicion, now that there has been a reduction in hospitalisation and deathly outcome after vaccination in some countries, this may affect the changes in attitudes towards vaccine acceptance [28,29].

This situation is different form everything we have faced before. There are a number of insufficiently known factors influencing the spread of COVID-19 infection such as unknown biology and mutagenic capacity of the causative agent, efficacy of new vaccines on current circulating strains, harmful vaccine side effects and vaccine efficacy in the event of altered virus properties. So decision to get vaccinated is more difficult than before.

In the case of COVID-19, there is a dynamical environment because of many uncertainties that exist and change over time. In order to provide right information and communication strategies to particular socio-demographic groups, a similar survey, as it is one presented in this manuscript, has to be repeated several times during the course of the pandemic.

The groups most at risk for developing serious complications of COVID-19 are similar to those at high risk for developing more severe consequences of seasonal flu—patients suffering from chronic diseases such as: diabetes mellitus, hypertension, heart and pulmonary diseases; and those over 65, as well as health professionals who are on the front line of defence against the virus [30]. Therefore, it is very important to vaccinate these groups of people. In our country, there is a low coverage with influenza vaccine, as there is no awareness among the healthcare policy makers on the importance of vaccination of healthcare workers against influence, and there are no strategies to improve their acceptance of the vaccine. This situation may be different in the case of COVID-19 pandemic—under the influence of the threat of the infection. This study was performed in the time frame when vaccine has become to be available to the general population and when individual persons have started to think about whether to receive it or not. The aim of the study was to examine the opinion of specific demographic groups towards COVID-19 vaccination, to detect which subgroups have which types of negative attitudes and reasons for refusing vaccination. We believe that such survey, if repeated in different time periods of the pandemic evolution, may help health workers and public health authorities in planning strategies to improve the vaccination coverage and to eliminate the possible misinformation that could affect vaccine hesitancy.

## 2. Materials and Methods

This study was conducted in facility in Osijek in December of 2020 and January 2021. Osijek is a city with 70,000 inhabitants and the administrative centre of eastern Croatia. The study was conducted on included 56 men and 220 women (*n* = 276), divided into three age groups: 20–40 years, 40–60 years and older than 60 years of age. Regarding their employment status, they were divided into groups: health professional, working in another profession, unemployed and retired.

Questionnaire form was used as a survey instrument and was distributed via e-mail, social media and instant messaging with free choice of participation. It is important to emphasise that the questionnaire form was conducted at a time when the vaccine was still being produced and it was not known when and whether it will come on the market and what the effectiveness of the vaccine will be. The questionnaire was divided into three groups of questions. The first group referred to information related to demographic and employment status: age group, gender, employment (health professional, working in another profession, unemployed and retired) and housing status (living alone, with partner without children/with minor or adult children, with children without partner).

The second group of questions was constructed to gather information on the perception of the dangers of COVID-19 infection, with the answers offered: yes, no and I don’t know. Questions were whether participants were considered susceptible to the disease or getting a more severe form of the disease, whether they were vaccinated regularly against seasonal flu and whether they received last year’s flu vaccine, and whether they planned to receive the COVID-19 vaccine.

The third group of questions provided information on the perception of the utility of the vaccine in relation to barriers, with answers offered: true, neither true nor false and false. This group was divided into three subgroups: organisational barriers (waiting in line for a vaccine, absence from work or obligations, availability of vaccine), perception of disease risk (severity of disease symptoms, re-infection after recovery) and perception of vaccine protective effect (level of awareness, safety and efficiency of the vaccine).

Responses were analysed multivariate according to age groups and categories of health and non-health workers.

Multiple regression analysis was applied to demonstrate the trend of the influence of individual factors on the decision to vaccinate against COVID-19. We used a multiple linear regression model and other statistical methods for statistical analysis: analysis of variance (two-way ANOVA) and calculation of the odds ratio to analyse the influence of different factors on vaccination decision in individual groups of subjects. The main purpose of statistical analysis in this paper is to show the relationship between variables represented by multiple linear regression (in our case with one dependent variable and several independent variables). The significance level for all statistical analyses was set at *p* = 0.05. The statistical software Past ver. 2.17c (PAST, Oslo, Norway) was used to analyse and evaluate statistical outcomes [31].

## 3. Results

Sociodemographic data of participants.

Most participants were under 60 years of age (75%). The reason for the predominance of the younger population could be that the surveys were conducted electronically and through social networks, which are mostly used by young people. More women (79.7%) than men (20.3%) participated which could also be explained with method of survey distribution because woman are more likely to participate in online research and social media activities [32,33]. Regarding their employment status most participants where employed (74.6%), mainly non-health care professionals (40.2%) and healthcare professionals (34.4%). Since most of the participants are from the younger population, they mostly live alone (29.3%), with a partner and minor children (30.8%) or just with a partner (24.6%). Characteristic are presented in Table 1.

2.Intention to get vaccinated

Woman make a significantly different decision depending on whether they are younger or older age group. More younger women decide not to get vaccinated, and older woman are more determined to get vaccinated (Figure 1). For men age is not of a major importance (Table 2).

Furthermore, employment status has a very important role in decision making. Non-healthcare workers are more prone to decide, compared to healthcare workers, not to get vaccinated, or they are in a greater part unsure of what to decide (Table 2, Figure 1).

3.Perception of one’s susceptibility for infection

Most participants, 52.6% of them, do not find themselves prone to getting diseases, while a slightly smaller number of them, 42.1%, are convinced they will not get a more severe form of disease.

Since the participants are mostly young, 65.2% of them do not suffer from any chronic illnesses, and those who do mostly have only one or two chronic illnesses.

Most of the participants, 74.4% of them, do not get seasonal influenza vaccine regularly, and did not get vaccinated this season. Characteristics are presented in Figure 2.

### 3.1. Perception of Disease Danger

Most participants are aware of the serious risk of contracting the disease, and the importance of vaccination. However, their opinion on the health risks is that they are not so susceptible to the disease and will get over it without more serious consequences, which probably affects their indecision regarding receiving the vaccine (Table 3).

Characteristic are presented in Table 3.

### 3.2. Perception of Vaccine Protective Effect and Risk Associated with Vaccine

There is a substantial level of uncertainty around the COVID-19 vaccine with the greatest distrust related to the potential side effects. Furthermore, there is no enough trust in vaccine efficacy.

Many participants are still undecided, they still form their opinion and think that they are not sufficiently informed to form strong views on the vaccine, which affects their decision.

The task for public health services is to provide information about safety of vaccine but problem in this unique situation is that knowledge regarding vaccine safety will change during the vaccination champaign and it is unlikely to be influenced by this factor before the vaccination campaign begins (Table 4 and Table 5).

### 3.3. Organisational Barriers

Regarding perception of vaccine utility in relation to organisational barriers, even though there is a substantial public concern about vaccine availability (Figure 3), this does not affect the vaccination decision (no significant correlation in a multiple regression model, not shown in the results).

## 4. Discussion

As there is an urgent need for a more updated and comprehensive understanding of attitudes towards vaccines, factors determining vaccine intent in the context of the COVID-19 pandemic should tailor public health messages accordingly. This study demonstrates difference in attitude toward vaccine in different socio-demographic groups (young-old, healthcare professionals-non-healthcare workers).

This study was conducted in time before vaccine was available in general population. A wide range of age groups was included, from 20 to over 60 years old. Even though most of the participants are from the younger age group, there is still a significant portion of risk groups among them, 34.8% of participants have one or more chronic diseases and 34.4% are a healthcare professional, for whom vaccination against COVID-19 is highly recommended (Table 1).

The decision not to vaccinate is influenced by age and employment status (Figure 1). Younger age groups and non-health workers showed greater hesitancy toward vaccine but also there was a lot of undecided, whose opinion is still forming. This suggest that different approaches are needed in the vaccination campaign. The target groups would be woman under the age of 40 years and non-healthcare workers because this groups are less willing to take vaccine (Figure 1). Furthermore, mental cognitive patterns regarding the relationship between the risk of disease consequences and perception of vaccine safety, influence the decision whether someone will take the vaccine or not (Table 3, Table 4 and Table 5).

The problem revealed by this research is that most of all participants still do not have a strong opinion about COVID-19 vaccination, mainly because they find that there is not enough information about the vaccine. Another problem is an otherwise poor propensity to vaccinate against seasonal flu, which is an optional vaccine that it is also recommended for at-risk groups (Figure 2).

Poor response to vaccination against seasonal flu among health care professionals has been a subject of much research due to increased risk of interpersonal transmission among staff and patients, which found that the main reasons for not getting the vaccine were fear of vaccine-related complications, lack of information and personal interest, not believing in vaccine efficacy or accessibility of the vaccine. On the contrary, employers willing to take influenza vaccine were motivated by a desire to protect themselves and patients or get vaccinated because employers require them to [34,35].

Zimmerman and coll. in their research found that the inconvenience of getting an influenza vaccine was one of the main reasons for not receiving the vaccine among healthcare professionals, especially if they cannot reconcile work commitments, and that offering benefits increases the vaccination coverage of this population [36]. Some studies have shown that using declination forms could also have some moderate effect on increasing the influenza vaccination rate [37].

Some of these reasons could be expected as potential reasons for non-vaccination against COVID-19 and can be prevented while some others, like vaccine related complications, and information regarding vaccine efficacy are unpredictable and are likely to change during pandemic.

Dror et al. in their study showed that vaccine acceptance among medical teams in COVID-19 departments is around 94%, which is higher compared to 75–91% in other medical departments. Furthermore, medical staff who think they have a higher risk of getting the disease are more susceptible to accepting the vaccine, and the most common fear is related to the question of vaccine safety [38]. This suggests that the decision of whether to vaccinate or not is influenced by personal experience and the type of information that is available. In this dynamic and unpredictable situation both factors are variable and the vaccination campaign needs to constantly adapt to these changes. Healthcare professionals have a key role in forming public opinion, so their attitude toward vaccine is of crucial importance [39]. The proportion of correspondents, in this research, who are certainly intending to get vaccinated with the COVID-19 vaccine is too low to achieve herd immunity since, according to the current evidence, achieving herd immunity takes between 55% and 82% of population [26].

Since this research was done before the vaccination started, in all subgroups (divided by age and by employment status) there are a lot of undecided respondents whose decision will likely to be formed based on internal and external factors, such as personal experiences or pieces of information received through public media or social networks.

In the context of Croatian culturological circumstances, as our experience teaches us, the source of information for elderly is mainly through national television and for the young also through social media.

The socio-economic situation in Croatia was not perfect before COVID-19 pandemic and it has even gotten worse since pandemic has started, mainly due to severe lock-down and preventive measures which resulted in increased unemployment and endangered existence of many people. Due to long-term epidemiological measures, also the quality of life has decreased [40]. Some public figures with a large number of fans and supporters on social networks made statement against vaccination and lock-down, with explanation that vaccination is chipping and that due to lock-down and cancellation of concerts and other curtail events they have suffered significant material damage.

According to the crisis committee reports on TV, the main causes of minor epidemic outbreaks, in Croatia, is a gathering of young people in cafes and clubs, which could also be a consequence of increased unemployment, as well as a gathering of people for religious and traditional customs. On the contrary, due to differences in the population density of individual counties in Croatia, there is a difference in the number of infected and epidemiological measures for individual counties, which leaves people with unanswered questions and with an impression of inconsistency.

Another problem, that is less likely to be influenced by the communication campaigns, is that, compared to well established vaccination protocols against infectious diseases found in the regular vaccination calendar, like measles, mumps, rubella and diphtheria or infectious diseases with well-established vaccination campaigns, like seasonal flu, specificity in COVID-19 pandemic and vaccination as a protective measure is that now there is still a lot of unknown about effectiveness and side effects of vaccine, and also there is a question whether pandemic could be stopped with vaccination [41]. Furthermore, there is a new technology of mRNA vaccine, which has not yet been in a massive use, so it is not known with certainty what the effect of this vaccine will be. However, this type of vaccine is expected to be more effective in elderly people, compared to vaccines that are produced by well-known technology of protein vaccinations, because mRNA vaccine bypasses the immune system sensibilisation step, which might be compromised in the aging immune system [42,43,44].

Furthermore, the first clinical trials for testing the efficacy of the novel COVID-19 vaccines were of a short-term, and the current on-going vaccination campaign is one large clinical test, where it is uncertain whether this vaccine will protect from new, mutated strain of virus [45,46].

It is important to follow recommendations of healthcare professionals and to adapt guidelines to the dynamic of the pandemic, by following the principle “watch and waiting”, and according to the characteristics of the specific socio-demographic groups. It is primarily necessary to protect the elderly and other most vulnerable groups, and when this is accomplished, the middle aged and younger become susceptible to infection.

When pandemic started, younger people were considered a less at-risk population for severe form of disease but lately, there are reported severe form of disease in young people without other comorbidities, as well [47,48,49,50]. In addition to a sense of conscience and solidarity with the population, this could be a powerful motivation for them to get vaccine. Many of this information was lacking in the time the survey was done which might have influenced the low motivation among the young population groups and non-healthcare workers for vaccination. This survey, if repeated, can help us understanding which socio-demographic groups are guided by which perception and attitudes towards vaccination.

According to all that, it is to expect that there is a hesitate of the public and also oscillations in the motivation for vaccination between the fear of vaccination and the fear of spreading the epidemic.

Strong argument of well-established vaccination campaigns, experience that vaccination has been a good protect against many dangerous infectious diseases cannot be used now when viral mutations and vaccine itself are still insufficiently known. It is necessary to monitor the course of the pandemic and results and effectiveness of the vaccination, with constant adaptation to the situation, which makes communication with the public very complex.

Research about COVID-19 vaccine acceptance in the general population and among healthcare professionals were conducted in many countries around the world with very variable results [51].

In the general public, by December 2020, the greatest acceptance of COVID-19 vaccine had Ecuador, Malaysia, Indonesia and China, with acceptance rates between 91% and 97%, and the lowest rates were found in Kuwait, where only 23.6% of the general population were intending to take COVID-19 vaccine [52,53,54,55,56].

Italy and France were not much better, with acceptance rates of 53.7% and 58.9%, respectively [57,58]. Furthermore, low rates of 54.9% were found in Russia, and between 56.9% and 75.4% in USA [58,59].

A recent study in the UK and USA demonstrated that as a consequence of disinformation, interest in vaccine has declined since September 2020 [60].

Research regarding healthcare professionals demonstrated the highest acceptance rates, of 78.1%, in Israel, and the lowest rates in the Democratic Republic of the Congo 27.7% [38,61].

Sallam M. in his research concluded that there is a need for more studies regarding the COVID-19 vaccine acceptance in Eastern Europe and this research is reporting the situation on this issue in Croatia [51].

A recent study about acceptance of the COVID-19 vaccine among healthcare professionals showed that the high risk of getting an infection at work is the main reason for willingness to get vaccinated, contrary to the general population where vaccine safety was the main factor [62].

In the preparation of strategies for vaccination campaigns, as our results also indicate, more knowledge about cognitive decision-making processes is needed and a greater application of knowledge from behavioural science. There are two basic strategies for influencing people to accept certain preventive measures: boosting (empowerment) and nudging (pushing them gentle) [63]. One strategy may be more appropriate, than the other, to tailor to certain social groups or to apply in certain situations. According to our results, boosting would be a strategy for motivating non-healthcare workers of middle of older age to get the COVID-19 vaccine, while more intensive “force” should be used to make a turn in the negative attitudes towards vaccination among younger women.

It is important to evaluate vaccine acceptance in specific socio-demographic groups in every country in order to better plan actions and campaigns to increase acceptance rates.

As we examined the perception of vaccine protective effect and risk associated with the vaccine, we need to conclude that, in Croatia, in the time this survey was done, there was still a substantial level of uncertainty about the COVID-19 vaccine, with the greatest distrust related to the potential side effects. Many participants were undecided, and their opinion was still forming (Figure 3). This uncertainty might have been related to concerns about the short time of the vaccine development, which according to the past experience usually lasts up to 10 years [62]. This fact should trigger more discussion among experts and motivate world authorities to better promote long-term vaccination strategies. Furthermore, there is an urgent need for a more up-to-date and meaningful understanding of vaccination factors and attitudes in the context of the COVID-19 pandemic, in order to tailor better public health messages. World Health Organization (WHO) suggests that there is a need for safe and effective vaccines to end the pandemic but for now vaccination does not mean that we can continue living lives like before pandemic [64]. Even though shrouded in controversy and numerous ethical and health issues, administrative restrictive measures on the free movement of the population, such as the introduction of EU regulations on the need to have COVID-19 passports, could change the decision to vaccinate, especially in the younger population [65].

The limitation of our study is that most of the correspondents were from the younger age group, the main reason being that the surveys were conducted electronically and through social networks, which are mostly used by young people. Despite this limitation, there was a significant proportion of high-risk correspondents, healthcare professionals and chronic patients. Furthermore, it is important to emphasise that our study was conducted when there was still not enough information on the COVID-19 vaccine and vaccination has not yet started fully, so it is possible that now when vaccination takes place all over the world with media reports, public opinion might have changed. We also highly support the Australian model of organisation of COVID-19 vaccination, which promotes a major roll of general practitioners and primary care physicians in educating and administration of the COVID-19 vaccine in the general population, as they have been confirmed as the most trusted source of information [63]. There is a pressing need for public health services to address the root causes through improved preventive health strategies using a wide range of policies, interventions and technologies.

## 5. Conclusions

Different demographic groups have different external and personal influences that are reflected in their cognitive processes. That is why it is necessary to adjust the way of informing each individual group about the vaccine. Vaccination of health professionals is especially important because of the devastating impact of staff shortages on the health system during the COVID-19 pandemic. Furthermore, this would reduce nosocomial transmission of the virus among the staff and patients. The goal of the campaign for the COVID-19 vaccine should be to raise awareness of the importance of the vaccine, including information on potential personal risks. Therefore, it is extremely important to provide an adequate level of education to health professionals and to provide the general public with access to accurate information related to the safety and efficacy of vaccines.

During a pandemic, depending on how effective the vaccine proves to be and on the dynamic of the spread of the infection, the attitudes of individual subgroups of the population may change, which is why similar surveys should be repeated several times during a pandemic. There is a need for carefully listening to the rumours of the public and critically analysing new evidence related to the COVID-19 outbreak, and according to it, for a fine-tunning the vaccination communication and motivation strategies.

## Figures and Tables

**Figure 1 ijerph-18-06141-f001:**
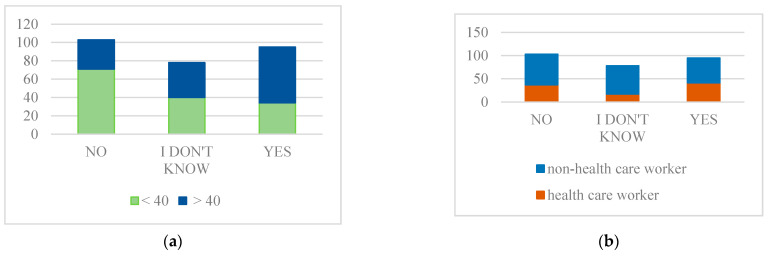
Intention to get vaccinated depending on age and the employment status. (**a**) More younger woman decide not to get vaccinated (**b**) Non-healthcare workers are more prone to decide not to get vaccinated. (Survey question: Are you planning to get vaccinated against COVID-19?).

**Figure 2 ijerph-18-06141-f002:**
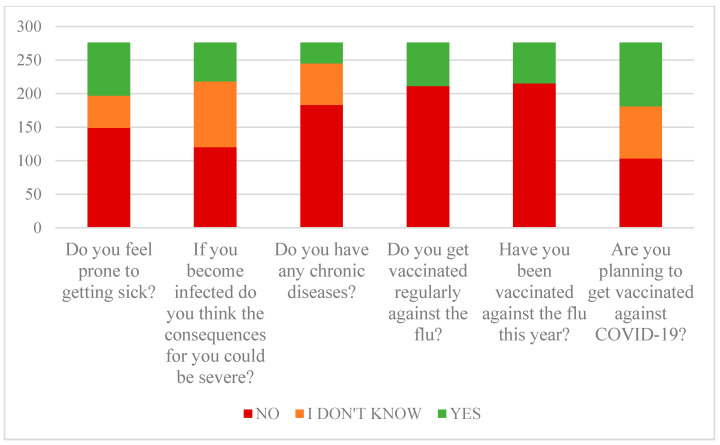
Perception of susceptibility to infection.

**Figure 3 ijerph-18-06141-f003:**
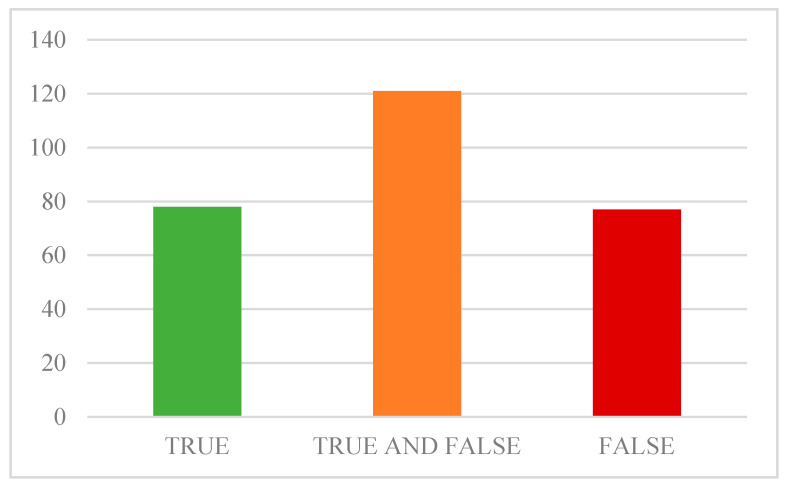
The vaccine will be easy to obtain.

**Table 1 ijerph-18-06141-t001:** Sociodemographic data of participants.

	Frequency	Percentage (%)
**Age:**		
20–40	145	52.5
40–60	62	22.5
>60	69	25.0
**Sex:**		
Male	56	20.3
Female	220	79.7
**Employment status:**		
Health care worker	95	34.4
Non-health care worker	111	40.2
Unemployed	28	10.1
Retired	42	15.2
**Household status:**		
Living alone	81	29.3
With a partner, no children	68	24.6
With a partner and minor children	85	30.8
With a partner and adult children	31	11.2
With children, no partner	11	4.0

**Table 2 ijerph-18-06141-t002:** The odds that a particular group of participants will not receive the vaccine.

Group	*z*-Value	OR *	*p*-Value †	CI ‡	
				5%	95%
<40 years of age	3.977	2.845	**<0.001**	1.7	4.763
>40 years of age	3.977	0.352	**<0.001**	0.21	0.588
Healthcare worker	2.202	0.560	**0.028**	0.334	0.938
Non-healthcare worker	2.202	1.786	**0.028**	1.066	2.992
Male < 40 years of age	0.957	1.7460	0.3387	0.558	5.469
Male > 40 years of age	0.957	0.5727	0.3387	0.183	1.794
Female < 40 years of age	3.98	3.241	**<0.001**	3.98	5.782
Female > 40 years of age	3.98	0.309	**<0.001**	0.173	0.551

* Odds Ratio (Altman, 1991). † Calculated according to Sheskin. 2004 (p. 542). ‡ 95% confidence interval (Altman, 1991). Significant *p*-values are bolded.

**Table 3 ijerph-18-06141-t003:** The influence of the examined factors on vaccination decision. Model: Perception of disease danger.

	Coefficient	Standard Error	*t*-Test	*p*-Value *	R^2^
Constant	1.167	0.039	29.730	<0.001	
Do you feel prone to getting sick?	0.078	0.034	2.296	**0.022**	0.060
If you become infected, do you think consequences could be severe?	0.118	0.043	2.720	**0.007**	0.068
Do you have any chronic diseases?	−0.154	0.051	−3.047	**0.003**	0.013
Do you get vaccinated against flu regularly?	0.125	0.096	1.300	0.195	0.089
Did you get this year’s flu vaccine?	0.301	0.099	3.052	**0.002**	0.109

*n* = 276 (adj. R^2^ = 0.165; std. error = 0.435). ANOVA (F = 11.868, *p* < 0.001). * Multivariate linear regression analysis. α < 0.05. Significant *p*-values are bolded.

**Table 4 ijerph-18-06141-t004:** The influence of the examined factors on vaccination decision. Model: perception of disease risk.

	Coefficient	Standard Error	*t*-Test	*p*-Value *	R^2^
Constant	0.480	0.140	3.415	0.001	
I don’t think I’m going to get sick even if I don’t get vaccinated.	0.031	0.036	0.857	0.392	0.117
I was already COVID-19 positive, so I will definitely not be positive again.	0.018	0.038	0.490	0.624	0.037
When the vaccine stops the pandemic, I won’t need to be vaccinated.	0.079	0.039	2.041	0.042	0.130
I don’t expect severe symptoms so there is no need to get the vaccine.	0.131	0.039	3.321	**0.001**	0.203
I don’t want to waste time on vaccination, I have more important worries.	−0.007	0.040	−0.176	**0.861**	0.085
I don’t care if I get sick.	−0.079	0.051	−1.547	0.123	0.026
I will not be vaccinated—it will be as it must be.	0.208	0.036	5.749	**<0.001**	0.262

*n* = 276 (adj. R^2^ = 0.317; std. error = 0.393). ANOVA (F = 19.204, *p* < 0.001). * Multivariate linear regression analysis. α < 0.05. Significant *p*-values are bolded.

**Table 5 ijerph-18-06141-t005:** The influence of the examined factors on vaccination decision. Model: perception of vaccine safety.

	Coefficient	Standard Error	*t*-Test	*p*-Value *	R^2^
Constant	1.784	0.115	15.524	<0.001	
I believe in the effectiveness of the vaccine.	−0.034	0.041	−0.811	0.418	0.214
I am afraid of the side effects of the vaccine.	0.048	0.029	1.684	0.093	0.077
I think the vaccine will protect me from getting sick.	−0.132	0.041	−3.220	**0.001**	0.247
I am not sufficiently informed about the vaccine.	0.089	0.028	3.190	**0.002**	0.104
I think COVID−19 vaccine is safe.	−0.171	0.039	−4.358	**<0.001**	0.288

*n* = 276 (adj. R^2^ = 0.364; std. error = 0.380). ANOVA (F = 32.448, *p* < 0.001). * Multivariate linear regression analysis. α < 0.05. Significant *p*-values are bolded.

## Data Availability

The data presented in this study are available on request from the corresponding author. The data are not publicly available due to data confidentiality.

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
