# Peer review of "Lack of Informations about COVID-19 Vaccine: From Implications to Intervention for Supporting Public Health Communications in COVID-19 Pandemic"

_ijerph, 2021, doi:10.3390/ijerph18116141_

Round 1

Reviewer 1 Report

This is a resubmission of the article. I appreciate that the authors have done a lot to improve its quality. However, the main problem still lies in the selection of the study sample. The corrections have not resolved or eliminated it. 
Lines 340-341 - how do we know this? What is the source of this belief?
The article does not take into account a number of factors that may influence attitudes towards vaccination, e.g. calls for the introduction of so-called covid passports (certificate) or the very likely need for a third dose of vaccine.
There is a concern that the data collected by the authors cannot fully explain the current reality. 
I must admit that I have serious doubts about the advisability of publishing this text. The argumentation is one-sided and biased, focused on the question: how to convince the unconvinced to accept the vaccine. Meanwhile, as one reviewer accurately noted in a previous round of reviews, "This is an unprecedented situation, both in terms of the spread of the pandemic and the number of experimental vaccines now available, so any parallels with previous mass vaccinations should be taken with great caution." 

Author Response

This is a resubmission of the article. I appreciate that the authors have done a lot to improve its quality. However, the main problem still lies in the selection of the study sample. The corrections have not resolved or eliminated it. 

Lines 340-341 - how do we know this? What is the source of this belief?

We erased this part.

The article does not take into account a number of factors that may influence attitudes towards vaccination, e.g. calls for the introduction of so-called covid passports (certificate) or the very likely need for a third dose of vaccine.

We have better explained and included all the factors that may influence the change of opinion about vaccination from the time the research was done to the present day when vaccine is available in general population.

We used the concepts of the Health Belief Theory to monitor the situation regarding self-perception of people in the general population towards COVID-19 vaccination, in the time point before the vaccination has started, and when the vaccine has been rapidly developing around the world. Also we paid attention on barriers to improved immunisation coverage in adults, focusing on  general attitudes toward health and vaccines, habit, awareness and knowledge. The public health workers have the task to understand the reasons for this hesitancy and to provide a support in making a decision.

There is a concern that the data collected by the authors cannot fully explain the current reality. 
I must admit that I have serious doubts about the advisability of publishing this text. The argumentation is one-sided and biased, focused on the question: how to convince the unconvinced to accept the vaccine. Meanwhile, as one reviewer accurately noted in a previous round of reviews, "This is an unprecedented situation, both in terms of the spread of the pandemic and the number of experimental vaccines now available, so any parallels with previous mass vaccinations should be taken with great caution." 

We emphasized that compared to well established vaccination protocols against infectious diseases found in the regular vaccination calendar, like measles, mumps, rubella and diphtheria, or infectious diseases with well-established vaccination campaigns, like seasonal flu, specificity in COVID 19 pandemic and vaccination as a protective measure is that now there is still a lot of unknown about effectiveness and side effects of vaccine, and also there is a question whether pandemic could be stopped with vaccination.

We refer to WHO which suggest benefit of being vaccinated in global meaning, particularly because research is still ongoing into how much vaccines protect not only against disease but also against infection and transmission. Also, as Covid 19  presents challenge in every dan practice, we have tried to  provide right information and communication strategies to particular socio-demographic groups, a similar survey, as it is one presented in this manuscript, has to be repeated several times during the course of the pandemic.

Also we emphasized that there is a new technology of mRNA vaccine, which has not yet been in a massive use, so it is not known with certainty what the effect of this vaccine will be. Although, this type of vaccine is expected to be more effective in elderly people, compared to vaccines that are produced by well-known technology of protein vaccinations, because mRNA vaccine bypasses the immune system sensibilization step, which might be compromised in the aging immune system.

Furthermore, the first clinical trials for testing the efficacy of the novel COVID-19 vaccines were of a short-term, and the current on-going vaccination campaign is one large clinical test, where it is uncertain whether this vaccine will protect from new, mutated strain of virus.

But also, there has been a reduction in hospitalisation and deathly outcome after vaccination in some countries, which may affect the changes in attitudes towards vaccine acceptance and also encourage medical staff and the general population to promote vaccination but with caution and individual approach. 

World health organization (WHO), on its official websites suggests, I quote:

“Equitable access to safe and effective vaccines is critical to ending the COVID-19 pandemic, so it is hugely encouraging to see so many vaccines proving and going into development. WHO is working tirelessly with partners to develop, manufacture and deploy safe and effective vaccines. 

Safe and effective vaccines are a game-changing tool: but for the foreseeable future we must continue wearing masks, cleaning our hands, ensuring good ventilation indoors, physically distancing and avoiding crowds. 

Being vaccinated does not mean that we can throw caution to the wind and put ourselves and others at risk, particularly because research is still ongoing into how much vaccines protect not only against disease but also against infection and transmission.

But it’s not vaccines that will stop the pandemic, it’s vaccination. “ 

Reviewer 2 Report

I would like to thank the authors for taking into consideration my requests for suggestions and improving the study in terms of analysis and research design. However, I feel that there are still some aspects that the authors should consider:

1) the authors base their arguments on the importance of achieving mass immunization through vaccines and compare COVID-19 vaccines to those for polio, rubella, etc. However, not only is there no data to confirm this assumption, but there are more and more cases of vaccinated people contracting the disease, sometimes in a severe form. Herd immunity is far from being confirmed. I suggest, therefore, that we use much more cautious tones and take a more neutral tone when presenting the reasons for the study. I believe the research would gain more from a less emphatic and 'politically' oriented approach if the authors simply reported the study as an exploratory investigation. I would also suggest avoiding terms such as 'conspiracy theories', which have a strong negative connotation. The aim of scientific research is to share results as neutrally as possible, not to make propaganda. 
2) I still do not see the connection between the survey and the role of social media, since the study does not analyse how the people in the sample formed their opinions. It is too easy to blame social media for misinformation and vaccine hesitation, this should also be proven. 

Author Response

  • the authors base their arguments on the importance of achieving mass immunization through vaccines and compare COVID-19 vaccines to those for polio, rubella, etc. However, not only is there no data to confirm this assumption, but there are more and more cases of vaccinated people contracting the disease, sometimes in a severe form. Herd immunity is far from being confirmed. I suggest, therefore, that we use much more cautious tones and take a more neutral tone when presenting the reasons for the study. I believe the research would gain more from a less emphatic and 'politically' oriented approach if the authors simply reported the study as an exploratory investigation. I would also suggest avoiding terms such as 'conspiracy theories', which have a strong negative connotation. The aim of scientific research is to share results as neutrally as possible, not to make propaganda. –

We emphasized that compared to well established vaccination protocols against infectious diseases found in the regular vaccination calendar, like measles, mumps, rubella and diphtheria, or infectious diseases with well-established vaccination campaigns, like seasonal flu, specificity in COVID 19 pandemic and vaccination as a protective measure is that now there is still a lot of unknown about effectiveness and side effects of vaccine, and also there is a question whether pandemic could be stopped with vaccination .

Also that there is a new technology of mRNA vaccine, which has not yet been in a massive use, so it is not known with certainty what the effect of this vaccine will be. Although, this type of vaccine is expected to be more effective in elderly people, compared to vaccines that are produced by well-known technology of protein vaccinations, because mRNA vaccine bypasses the immune system sensibilization step, which might be compromised in the aging immune system.

Furthermore, the first clinical trials for testing the efficacy of the novel COVID-19 vaccines were of a short-term, and the current on-going vaccination campaign is one large clinical test, where it is uncertain whether this vaccine will protect from new, mutated strain of virus.

2) I still do not see the connection between the survey and the role of social media, since the study does not analyse how the people in the sample formed their opinions. It is too easy to blame social media for misinformation and vaccine hesitation, this should also be proven. 

As it is known from previous reports that there have been multiple factors identified that influence immunisation uptake among adults including social influences, disease-related and vaccine-related factors, general attitudes toward health and vaccines, habit, awareness and knowledge, practical barriers and motivators, and altruism, we have used the concepts of the Health Belief Theory to monitor the situation regarding self-perception of people in the general population towards COVID-19 vaccination, in the time point before the vaccination has started, and when the vaccine has been rapidly developing around the world.

The Health Belief Theory helps us understanding the reasons why some socio-demographic groups resist to receive some preventive measures, including the hesitancy towards vaccination, by revealing the relationships between the self-perceived costs and benefits if one decides to receive some preventive measure (i.e. vaccine). 

Although the situation with COVID 19 pandemic is specific and different from previous needs for mass vaccination, in terms of the spread of the pandemic all over the globe, many undetermined factors that influence this spread, and the number of experimantal vaccines that are now available – the public concern regarding vaccination is always about usefulness and safety of vaccines.

Also The World Health Organization (WHO) emphasized that the outbreak of COVID-19 and the response measures are accompanied by abundant information, and it is difficult to find reliable sources and reliable guidance in order to coordinate the search for sources, identify, and reduce false information spread.To conclude with, we support idea of more comprehensive approach of understanding of COVID 19 and immunisation process.  On the other hand  there is a dynamical environment process because of many uncertainties that exist and change over time. In order to provide right information and communication strategies to particular socio-demographic groups, a similar survey, as it is one presented in this manuscript, has to be repeated several times during the course of the pandemic.

Reviewer 3 Report

Thank you for making changes.

Author Response

We made corrections of sentences and English correction.

Round 2

Reviewer 1 Report

I admire the authors' determination to get their text published. Despite many reservations, I have to admit that they did a lot to improve its quality. This does not change the fact that we must approach the processes we are witnessing with great humility. Scientific journals must be places where opinions, theories, hypotheses, and research results confront each other. Therefore, the reviewed article can be regarded as a voice in this discussion. It is a rather one-sided voice and not entirely methodologically convincing (the sample!), but I think it can be allowed to resonate in the space of scientific dispute.

Reviewer 2 Report

I would like to thank the authors for considering my last remarks and I appreciate the new statements that have been added. I find that the study is much more balanced now as it includes more cautious and circumspect claims. I only point out some spelling problems, such as the ones indicated below:

"neccessary", "position od constantly", "This situation is different form everything", "influenca vaccine", "carefully leastening".

This manuscript is a resubmission of an earlier submission. The following is a list of the peer review reports and author responses from that submission.

Round 1

Reviewer 1 Report

1. in some footnote number brackets, there is an unnecessary space before or after the footnote number. Unnecessary spaces are also before periods (e.g., line 40). This should be carefully edited out.
2. the sentence in lines 46-49. The three publications cited are from 2001, 2010, 2013. Since then, both the anti-vaccine movement and scientific reflection on the phenomenon have deepened considerably. It would be helpful to cite more recent literature and expand this thread a bit, as it is quite relevant to the COVID-19 vaccine discussion. It would certainly be helpful to deepen the Discussion and Conclusions. In the following footnotes (up to No. 14) more recent publications appear, the youngest is from 2016. A bit more references to publications such as in footnote No. 15 would be useful. A point to the role of publications in so-called predatory journals as a source of argumentation of the anti-vaccine movements is to be considered (although this is a thread perhaps too digressive to the main topic of the article).
3. the paragraph from line 59 to 101 is too long and contains too many threads. It should be divided.
4. the objectives of the article are defined very perfunctorily (98-101). The way they are defined places the article more in the field of management or communication sciences or political sciences, but the authors did not define it clearly.
5. Perhaps the most serious doubts concern the selection of the sample. If we take 70,000 as the population size - then a sample of 276 people gives us a possible error of 6%. This is a lot for a quantitative study. Moreover, the sample is not representative. It overrepresents certain groups. This makes the results almost worthless. They cannot be generalized to the entire population. I don't understand the idea of conducting quantitative research on such an imperfect sample. It would have been better to select some of these people and conduct qualitative research, for example in the form of an Individual In-Depth Interview. 
6. In my opinion, the questions contained in the questionnaire concern too obvious matters. It would be much more interesting to list the viewpoints of the surveyed group and to indicate where their knowledge about vaccines comes from in the first place (i.e. what is the main source of their knowledge).
7 I don't think that the results of the study bring any novelty to the current State of Art. Rather, they are quite obvious. Therefore (given the objections in points 5 and 6) I do not see the sense of publishing this article. The only way to salvage it would be to add the results of qualitative studies, taking into account the diversity of sources of information on vaccines and indicating possible future research.

Reviewer 2 Report

Dear authors,

thank you for submitting your study to Int. J. Environ. Res. Public Health. I found the manuscript interesting and it gave me a lot of food for thought which I would like to share with you.

To date, COVID-19 vaccines have been authorised from the European Commission following evaluation by the European Medicines Agency (EMA), but they have not been approved. For example, on January EMA has recommended granting a conditional marketing authorisation for COVID-19 Vaccine Moderna. This means that any similarity with vaccines developed to contrast polio, measles, rabies, typhus is incorrect, as COVID-19 vaccines have not been fully tested yet. On the contrary, what we are witnessing is the third phase of the experimentation with large numbers, with new data about adverse effects collected. This has lead, for instance, to the temporary halt of AstraZeneca (aka Vaxzevria) in many European countries. The situation regarding the benefits and even important adverse effects (e.g. risk of thrombosis or blood clots, etc.) is far from clear, and it is entirely reasonable for people to feel frightened and disoriented since it is not possible to provide them with definite and incontrovertible information. From this point of view, the statement that "Distrust in the efficacy and safety of the vaccine, emphasize the need for health services to provide better information to the general population" is simply unsustainable, as there is no definitive information on the safety of these vaccines. Moreover, it is becoming increasingly clear that these vaccines do not confer immunisation, are at best a temporary cover, and that the long and medium-term effects, especially of mRNA vaccines, are unknown.

I strongly disagree with statements such as "encouraging [social media users] to participate more actively in medical decision-making, this becomes an alarming problem with potentially dangerous consequences for the public". I fully understand that questioning medical authority, which until now has been unquestioned and assumed to be absolute truth, is not pleasant for those working in the health sector, but social media imply active participation in the many aspects of people's lives, from relational, political and civic aspects to the sphere of health. Instead of always accusing social media of spreading fake news and misinformation, academics could finally highlight its potential as a tool to make people more aware even of something as important as health. Medicine is not an objective science and it is normal for people to ask questions and express mistrust and perplexity on an issue as important as the vaccines for COVID-19, which, I repeat, have only been authorised for sale and are still in the massive testing phase. Moreover, against this backdrop of great uncertainty, it has been decided in Italy that health professions must be vaccinated against COVID-19, regardless of their age or state of health. We cannot say that this is not conducive to increasing public confidence in vaccines.  

In this light, while I appreciate the aim of your study, “to detect potential fears and reasons for refusing vaccination in order to work on strategies to eliminate possible misinformation that could affect vaccine hesitancy”, I also invite you to consider the many reasons for which fears and hesitancy are currently more than justified. This is an unprecedented situation, both in terms of the spread of the pandemic and the amount of experimental vaccines now available, so any parallels with previous mass vaccinations should be taken with great caution.

In terms of methodological and technical problems of the study, I suggest addressing the following issues:

  • I advise reporting in the Abstract the number of participants involved (N=276).
  • Social media use was not a variable considered in the study (e.g., to investigate how misinformation about COVID-19 vaccines is spread), but only one of the channels through which the survey was distributed. I wonder for what reasons the authors have targeted social media in the Introduction.
  • The abstract states that "Ordinary least squares regression analyses were used in our study to examine the data”, however there is no trace of this analysis in the study. At the end of section 2. I found that “Responses were analysed multivariate according to age groups and categories of health and non-health workers”. First, I expected more justification for comparing results obtained from these two groups and more elaboration, either theoretically and methodologically. Do you expect to find different attitudes from the health workers? If you do, you also need to present arguments for possible differences in the first section (Introduction). Secondly, I do not see any multivariate analysis but only descriptive statistics (i.e., frequency and percentages), or t-tests to compare the two groups. I suggest to use some inferential model if you want to derive meaningful results from the study.
  • The Discussion contains many information that should be presented in the Introduction or in a Related literature section (e.g., “Poor response to vaccination against seasonal flu among health care professionals…”). Overall, this section does not interpret or discuss the results coherently, introduces too much new information and does not guide the reader to grasp the new knowledge added by this study. I recommend an extensive reorganization of the manuscript and a clearer focus on the research aims. Specifically, the study should fill a gap in current literature beyond the specific case study.

Reviewer 3 Report

Manuscript needs to be edited, as there are many small errors throughout. Also, the manuscript also needs to be written in past tense.

The participant section needs to justify the gender difference; there are many more women that were recruited.  Why?

What type of statistical program was used, what type of analysis?  What limitations?  I am almost wondering if you should just remove the men in this study and focus on women- that would make it certainly more novel and interesting. But, either way, you need to highlight and include more depth on your methodology.

Your discussion section doesn't flow- it's just blocky information.  And it could be enhanced by adding a cultural component, since this study occurred in Croatia.  How does this population differ from an African setting, western European, American?

Most of this information is already known.  Focus on what is novel about this study.  I would focus on gender or perhaps the Croatian population.  What is different, what can be learned, and what can be generalizable or transferable other populations in the world.